# Systematic review and theoretical comparison of children's outcomes in post-separation living arrangements

**Laura M. Vowels**[1]*, **Chiara L. Comolli**[2], **Laura Bernardi**[3], **Daniela Chacón-Mendoza**[4], **Joëlle Darwiche**[1]

**1** Family and Development Research Center (FADO), Institute of Psychology, University of Lausanne, Lausanne, Switzerland, **2** Department of Statistical Sciences, University of Bologna, Bologna, Italy, **3** LIVES, Institute of Social Sciences, University of Lausanne, Lausanne, Switzerland, **4** Center for Research and Political Studies, University of Costa Rica, San José, Costa Rica

* laura.vowels@unil.ch

## Abstract

The purpose of the systematic review was to synthesize the literature on children's outcomes across different living arrangements (nuclear families, shared physical custody [SPC], lone physical custody [LPC]) by extracting and structuring relevant theoretical hypotheses (selection, instability, fewer resources, and stressful mobility) and comparing the empirical findings against these hypotheses. Following the PRISMA guidelines, the review included 39 studies conducted between January 2010-December 2022 and compared the living arrangements across five domains of children's outcomes: emotional, behavioral, relational, physical, and educational. The results showed that children's outcomes were the best in nuclear families but in 75% of the studies children in SPC arrangements had equal outcomes. Children in LPC tended to report the worst outcomes. When compared with the different theoretical hypotheses, the results were the most consistent with fewer resources hypothesis which suggests that children especially in LPC families have fewer relational and economic resources whereas children in SPC families are better able to maintain resources from both parents.

## Introduction

High rates of union (cohabitation or marriage) dissolution have contributed to the increased number of children growing up with biological parents residing in separate households over the last decades [1]. The majority of prior research shows that divorce has a negative impact on children's outcomes [2–5] and that children who live in a shared physical custody (SPC; i.e., live with both parents between 30–70% of the time) perform better than children in lone physical custody (LPC; i.e., children live with only one parent) (for review, see Nielsen, 2018 [6]). However, the literature is divided on whether the differences in children's outcomes are due to the living arrangement itself or whether there are other factors that better explain the discrepancy in outcomes.

**Data Availability Statement:** The manuscript reports results from the systematic review and thus no original data are reported. Materials can be

found on the OSF project repository (https://osf.io/9w5rp/).

**Funding:** This study received funding "Seeds of LIVES" from the Interdisciplinary Research Centre LIVES, University of Lausanne, Switzerland paid to JD. The funders had no role in study design, data collection and analysis, decision to publish, or preparation of the manuscript.

**Competing interests:** The authors have declared that no competing interests exist.

Several studies have tried to distil factors that may explain the differences in children's outcomes across living arrangements. Some of these factors include: socioeconomic status [4, 7–9], interparental conflict [10, 11], parenting and parental factors [11], and parent-child relationship factors [12–14]. Prior research has also shown that in forerunner countries where SPC was legally implemented early on either as the preferred arrangement, such as in many Nordic countries and some parts of North America [15], the prevalence of SPC tends to be higher and be less strongly associated with parents' socio-economic status (SES) or conflict between them [16, 17]. On the contrary, in low-prevalence countries with little legal support for SPC (such as Germany [18]), separated parents opting for SPC tend to be more affluent, highly educated, less conflict-ridden, and hold more egalitarian beliefs about gender roles compared to LPC parents [6]. These studies seem to suggest that it is not the living arrangement itself that predicts children's outcomes but rather that there are other explanatory factors for the differences.

Several previous reviews and meta-analyses have been conducted in this topic with somewhat different focuses. Several of these reviews have focused primarily on the differences between SPC and LPC families showing that, overall, children tend to do better in SPC than in LPC families [6, 19, 20]. For example, Nielsen's (2018) [6] review of around 60 studies found that children in LPC fared worse than children in SPC across five broad domains: academic or cognitive outcomes, emotional or psychological outcomes, behavioral problems, overall physical health or stress-related physical problems, and the parent–child relationship quality. They also examined interparental conflict and income as potential explanatory variables of the difference between SPC and LPC. However, it is not clear from the review whether it was systematic and what time period the studies were included from. We also expand on this review in several ways: by extracting key hypotheses from the existing theories in the field and comparing the empirical findings against these hypotheses, by including nuclear families (i.e., families that constitute parents with their own biological or adopted children) in the comparison, and by comparing a larger number of possible explanatory variables.

Further, Jensen and Sanner (2021) [21] performed a scoping review of the methodologies used in studies examining children's living arrangements. Their review compared the studies based on how family structure was measured, what methods and analysis approaches were used, whether the studies were theory-driven or not, and how the studies conceptualized child well-being. The review provides a methodological foundation for the field which can inspire high quality future research. The present review incorporates Jensen and Sanner's critique of a lack of theoretical foundation in previous literature and goes further by confronting the theoretical hypotheses against the empirical results for nuclear, SPC, and LPC families.

Jensen and Sanner (2021) [21] noted that much of the literature is atheoretical meaning studies which did not explicitly identify a theory or a perspective that was used to frame their research questions and/or hypotheses regarding differences in children's outcomes across living arrangements. Without a theoretical basis, it is difficult to evaluate the choices made during the research process which may affect the results and contribute to the heterogeneity in outcomes across studies. Theories should provide a foundation for hypotheses, which living arrangements are compared, which explanatory variables (e.g., age, SES, interparental conflict, parent-child relationship quality) are included in the analyses, and why certain outcomes are chosen over others. When these assumptions are not made explicit, this affects the interpretation of the findings. For example, if authors hypothesize that separation is associated with worse outcomes for children than staying married, they could test this hypothesis in a myriad of ways including many different explanatory variables, different custody arrangements, and conceptualizations of outcomes which would potentially yield very different results depending on the choices made. In contrast, the hypothesis becomes more testable and the results more readily interpretable when researchers clearly specify their theoretical assumptions, for

example, by stating that separation is associated with worse emotional outcomes for children than staying married, but only if the child loses contact with one parent and only if the remaining parent has low income (fewer resources hypothesis derived from the social capital perspective [22, 23]. In the latter case, it is much easier for readers to evaluate whether the results support the hypothesis, and under which circumstances the results are expected to hold.

Despite the lack of explicit theories within papers, a myriad of theories exists that make different implicit or explicit predictions on how living arrangements affect children's outcomes and why. Thus, we begin our review by providing an overview of the existing theories and hypotheses that can be derived from these theories and then use the systematic review to examine whether there is support for these theories in the literature. Specifically, in Section 1, we deliver an overview of the existing theories, hypotheses, and explanatory variables that can be derived from these theories. We hope that this will provide guidance for researchers designing their studies and encourage a consideration and explicit reporting of guiding theories and assumptions. In Section 2, we identify whether there are systematic differences across children's living arrangements and compare the results in view of different theoretical hypotheses to better understand whether certain theories are supported by the literature. In Section 2, we also investigate whether the results differ based on a) the outcomes and b) the explanatory variables that were examined across studies.

## Section 1: An overview of the main theoretical hypotheses predicting child well-being in different living arrangements

In this section, we provide an overview of the main theoretical hypotheses from the literature, theories that fit within these hypotheses, and explanatory variables based on these hypotheses that may explain differences in children's outcomes across living arrangements. We included the most relevant theories within psychology and sociology that explain differences across children's living arrangements, most of which were also present in the studies included in the systematic review. We organized the theories based on the main hypotheses. In some cases, there were several additional hypotheses supporting the same outcomes in different living arrangements and theories that fit within the main hypothesis. Table 1 and the proceeding subsections are organized by four main hypotheses with other relevant hypotheses grouped within the same section where relevant: selection (differences in living arrangements are due to selection into these arrangements), instability (divorce and separation cause instability), fewer resources (divorced, especially LPC, families have fewer resources), and stressful mobility hypothesis (stress of movement and ruptures in relationships). The main hypotheses have been proposed in the literature but the grouping of existing theories into the hypotheses was made by the research team based on the theories' central assumptions. Similarly, the specific hypothesis regarding the order of the quality of living arrangements for children's outcomes was often inferred by the researchers rather than explicit in the theories. Finally, we use the term explanatory variables to refer to any variables that theories suggest can explain differences across living arrangements regardless of whether they are considered control, moderator, or mediator variables as the role of the explanatory variables was often not explicit in the theories. Explanatory variables were divided into the following categories: selection (pre-divorce parent and child characteristics), relationships (parent-child relationship quality, parenting time, parenting), economic (e.g., SES, income, parents' occupation), child (age, gender), and other (e.g., child's outcomes, previous time, transitions). We also included living with a stepparent as a potential explanatory variable.

### 1. Selection hypothesis: Differences in living arrangements are due to selection

The selection hypotheses [24–28] states that children in separated and non-separated families differ because of the family characteristics that precede separation. The hypothesis suggests

**Table 1. Summary of the main hypotheses, theories, and explanatory variables predicted by the theories related to child living arrangements and child well-being.**

| Hypothesis | Theories | Explanatory |
|---|---|---|
| **IF = SPC = LPC** | | |
| **Selection hypothesis** | | selection |
| Context hypothesis | Bioecological model | relationships<br>parents' well-being<br>child<br>economic |
| **IF>SPC≥LPC** | | |
| **Instability hypothesis** | Family instability | number of transitions (other)<br>stepparent |
| | Escape from stress theory | parents' well-being<br>relationships<br>stepparent |
| **IF≥SPC>LPC** | | |
| **Fewer resources hypothesis** | Parental loss theory | relationships<br>economic |
| | Social capital perspective | relationships<br>economic |
| | Economic deprivation | economic |
| | Economic strain perspective | relationships<br>economic |
| | Attachment theory | relationships |
| **IF≥LPC>SPC** | | |
| **Stressful mobility hypothesis** | Child's agency theories | number of moves between homes (other) |
| Role/boundaries hypothesis | Structural family theory | relationships |
| Primary attachment figure | Attachment theory | relationships<br>number of moves between homes (other) |

*Note*. IF = nuclear family, SPC = shared physical custody, LPC = lone physical custody.

that it is not the family living arrangement after separation that makes children in these families more vulnerable but rather that the separation itself is a consequence of their already existing higher vulnerability. Previous literature suggests that non-nuclear families, already before separation, are less wealthy and less educated, tend to display lower quality of parent-child relationship, greater levels of conflict, lower co-parenting, and investment in children [24–28]. Furthermore, income, conflict, parent-child relationships, and quality of parenting skills are the factors most frequently proffered as the reasons why SPC children have better outcomes than LPC children [6, 29]. Based on the selection hypothesis, we expect no significant differences across the living arrangements after accounting for selection variables in the analyses.

In addition to the selection hypothesis, the context hypothesis (based on the bioecological model) also suggests that it is not the living arrangements themselves that affect children's outcomes, but it is the context within which the children live. The context hypothesis suggests that it is the overall context (e.g., conflict between family members, social environment) that affects children's outcomes rather than living arrangements alone. Thus, it also assumes that differences in living arrangements are due to selection. Therefore, these two hypotheses are combined in Table 1 under selection hypothesis. Based on the context hypothesis, other possible explanatory variables include relationships (e.g., parent-child relationship quality, conflict between parents), parent's well-being, child variables, and economic situation as these form a part of the bioecological system.

## 2. Instability hypothesis: Divorce causes instability regardless of subsequent living arrangement

The instability hypothesis predicts a negative effect of union dissolution on children's outcomes because of the detrimental effect of family instability ([30]; see also escape from stress theory, [2, 31, 32]. The hypothesis predicts that children fare worse the greater the number of changes in family structure they experience. Based on this hypothesis, children in nuclear families should have better outcomes compared to children in non-nuclear families regardless of the subsequent living arrangement. However, having a stepparent would include another transition, and as such it would be expected that stepparent families fare worse compared to all other living arrangements.

The escape from stress theory [2, 31, 32] also supports the instability hypothesis. However, it suggests that because of the stress of union disruption, the strains of solo parenting, and the reduced financial means, single parents spend less time with their children, are less emotionally supportive, and report more conflict with their children than continuously married parents [31]. This theory points out that LPC may be associated with higher stress and, therefore, reduce parents' emotional availability to their children [2]. In stepfamilies, tensions between children and stepparents are not uncommon [31], especially when parents and stepparents have poor marital relationships [33, 34]. Thus, the escape from stress theory would also predict that SPC families do better than LPC families.

The expected explanatory variables based on the instability hypothesis for the differences between living arrangements would include the number of transitions as well as having a stepparent. Additionally, the escape from stress theory would also suggest that the differences may be due to a single parent's poorer well-being or poorer parent-child relationship.

## 3. Fewer resources hypothesis: LPC families have fewer resources compared to other families

The fewer resources hypothesis [35] suggests that the reason children from divorced families fare worse than children in nuclear families is because they lose economic or relational resources. Children in LPC arrangements may be particularly prone to fewer resources as they may lose access to one parent. It is generally easier to maintain close contacts and access to both parents' economic and relational resources in SPC arrangements [36], however, SPC families may be lower in economic resources due to the costs associated with running two households compared to nuclear families.

There are several theories that fit within this hypothesis. For example, the parental loss theory [26, 35] and the social capital perspective [22, 23] suggest that poor children's outcomes associated with the absence of one biological parent may be related to reductions in parenting time, finances, parental energy, and the lower social and human capital remaining in the household to devote to childrearing. Similarly, attachment theory posits that maintaining multiple attachment figures in child development and caretaking in multiple contexts promotes each parent's bond with the child which is beneficial for the child [37, 38]. Thus, children in SPC families would have more resources compared to children in LPC. Moreover, economic deprivation [32] and economic strain theories [2] focus solely on the economic loss caused by divorce.

These theories argue that children in LPC arrangements, in which they live primarily or exclusively with one of their parents and have either no or only limited contact with their other parent [39], generally suffer from a significant loss of financial resources previously provided by their non-residential parent [40]. Overall, these theories conclude that parental separation and loss of contact with the non-resident parent implies a reduction of emotional, relational, and financial resources important to the child, which explains the lower well-being

and slower development of children in LPC arrangements compared to other living arrangements. For example, a wealthier single parent may have more resources than lower-income SPC families and maintaining parental contact with the non-resident parent even in LPC arrangements may mitigate the potential negative association of these family structure with children outcomes. Thus, we expect that economic and/or relationship variables will explain at least some of the differences between living arrangements.

## 4. Stressful mobility hypothesis: SPC results in stressful movement between households and potential ruptures in relationships

The stressful mobility hypothesis assumes that having to move between households (potential explanatory variable) can be stressful for children with more frequent moves adding to the stress [41, 42], and that the level of stress due to mobility can vary depending on the age of the child and the level of conflict between parents [36]. In one study [36], older children reported higher stress when moving between households compared to younger children. This may be, for example, because the moves cause disruption in their friendships or out-of-school activities. They also showed that children in higher-conflict families showed higher levels of stress in SPC arrangements compared to children in low-conflict families.

Family systems theories [22, 43–45] and attachment theory [46], in turn, suggest that stability, clarity, and continuity of relationships are important for well-functioning families. For example, the family systems theories suggest that a healthy family structure is one in which there are clear, hierarchical boundaries while diffused or poorly defined boundaries characterize dysfunctional families [47]. When children experience frequent changes in the family environment, such as in SPC arrangements, this might be associated with more opportunities for conflict between parents and thus a higher risk of child triangulation and in turn poorer child outcomes. Furthermore, the primary attachment figure hypothesis of the attachment theory suggests that infants and young children should spend limited time away from their primary attachment figure to promote a secure attachment and limit the risk posed by attachment insecurity for children's future mental health [48]. Thus, the stressful mobility hypothesis suggests a clear superiority of nuclear over non-nuclear families. However, based on this hypothesis we would expect that LPC arrangements would be better for children's outcomes and would allow them to maintain a close relationship with their primary attachment figure without blurring the family boundaries and thus LPC should be preferable to SPC. Potential explanatory variables based on this theory would include the number of moves between homes and relationship quality between the parents as well as between parents and their children.

## 5. Conclusion

In summary, the four main hypotheses (selection hypothesis, instability hypothesis, fewer resources hypothesis, and stressful mobility hypothesis) that are supported by several major theories within psychology and sociology suggest differing outcomes across living arrangements. They also suggest several possible explanatory variables that may explain the differences in children's outcomes across living arrangements. In the systematic review presented in the section below, we aim to compare the empirical findings against these hypotheses.

## Section 2: The systematic review

### Methods

For the current systematic review, we reviewed existing empirical social sciences literature from the beginning of 2010 to the beginning of December 2022. The systematic review

followed the implementation of the guidelines from the Preferred Reporting Items for Systematic Reviews and Meta-Analysis (PRISMA). The systematic review was not registered on PROSPERO but all the material pertaining to the review including the protocol as well as all supplemental files can be found on the Open Science Framework: https://osf.io/9w5rp/.

**Inclusion and exclusion criteria.** The inclusion and exclusion were based on the PICO (Patient/Problem, Intervention, Comparison, and Outcome) components:

1. The publications had to focus on divorce or separation either of cohabiting or married couples.

2. They had to include outcomes for children and young people aged between 0–18 years old given custody arrangements would still apply to them.

3. They had to include a comparison with nuclear families, SPC, and LPC. SPC was defined as the child living with each parent between 30–70% of the time. LPC was defined as the child living mostly or solely with one parent.

4. The studies had to examine the effects on children's outcomes broadly defined to include academic or cognitive outcomes, emotional or psychological outcomes, behavioral problems, overall physical health or stress-related physical problems, and the parent–child relationship quality.

5. We concentrated on population(s) or context(s) from Western European countries, North American countries, Australia, and New Zealand where divorce and shared physical custody have become increasingly common.

6. We chose to include papers from January 2010 onwards given SPC arrangements have become increasingly common and warrant an evaluation considering theoretical hypotheses in the literature.

7. We only included articles which were written in English and published in a peer reviewed journal.

**Search strategy.** We chose three well-known electronic bibliographic databases (i.e., Scopus, Web of Science Core Collection, and APA PsycInfo) that contain a large catalog of peer-reviewed journals and publications from a variety of disciplines and research fields. For each of the databases, after several initial test searches and consequent adjustments, a final search strategy was elaborated using specific entry words associated with the PICO components parameters and eligibility criteria. The search strategies were constructed with the use of a set of entry words and Boolean characters that were adjusted to the database's query language and requirements. With the defined search strategy, the final information search was conducted on 1 December 2022. We specified the publication period (2010–2022) and, when possible, country filters were selected, as well as language, discipline, and type of document. The searches were conducted with the advanced query option, and the exploration of the entry words was done on the title, abstract, and keywords of the indexed publications. The final number of records retrieved was 4,943. Basic information on the records from each database was extracted to spreadsheet files using the export available function and combined on a single dataset. We obtained information about authors, title, abstract, year, DOI, and source.

**Screening and selection.** The general process of screening and selection took place in the dataset document which compiled all the retrieved records that were organized with significant descriptive information, the indication of its inclusion or exclusion at three screening rounds, and a general specification of the reasons accounting for its exclusion. These screening

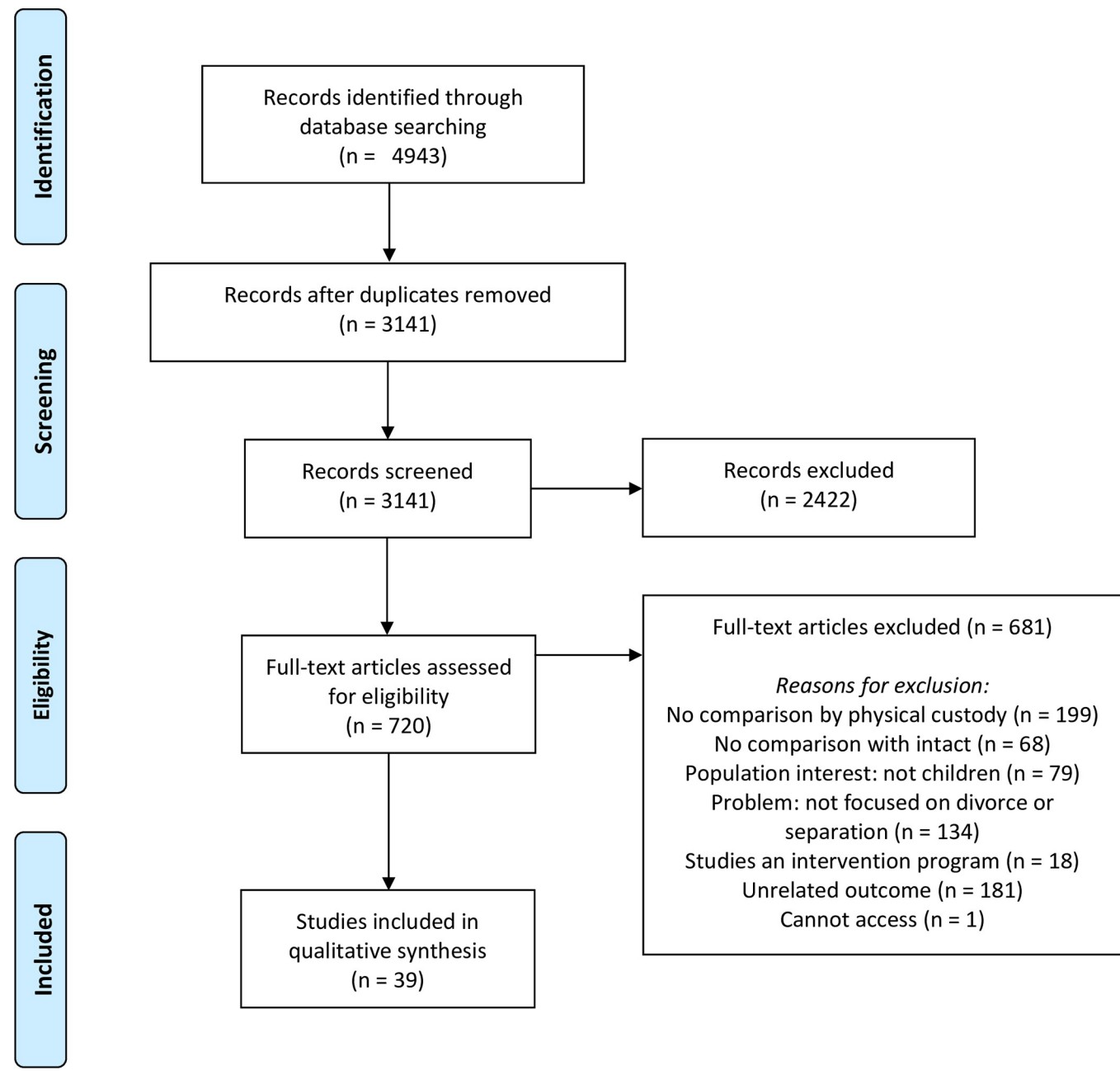

**Fig 1. PRISMA flow diagram.**

and selection steps are summarized in a flow-diagram in Fig 1. In the first round of screening, we identified and removed all duplicates ($n = 1,802$), this selection process was based on the information about authors, article title, and year of publication. The second round of screening was conducted to remove studies that based on the title and abstract did not fit the eligibility criteria. Additional exclusions were made based on the type of document (e.g., editorials, literature reviews, and commentaries not filtered during the information search). A total of 2,422 studies were removed. For the third round, we retrieved the full text of records included in the second round (i.e., 720 records), and created a separate data table to extract relevant

information to assess the selection of records based on the eligibility criteria defined for each component of interest (e.g., problem, population, context, outcomes). DCM, CLC, and LMV conducted the initial selection based on titles and abstracts, DCM and LMV read all full texts for the final selection of articles. When unclear, we consulted and resolved the selection with the principal researchers. Some studies have used the same dataset and thus were combined in the analyses. The final sample included 39 records. For a full list, see Table 2.

**Data extraction.** For the final included publications screened in the third round, we added and systematized information about other characteristics of importance for subsequent analysis (e.g., age range, net sample size, independent and dependent variables). We processed complex information regarding living arrangements and children's outcomes in a dichotomous format. We also generated pivot tables that summarized the data corresponding to categories of information and the processes of inclusion and exclusion of records. The outcomes were split into five categories following Nielsen's (2018) [6] categorization: emotional or psychological outcomes (e.g., feeling depressed, anxious, or having low self-esteem); behavioral problems (e.g., misbehaving at home or school, hyperactivity, and teenage substance use); the quality of parent–child relationships (e.g., communication with and closeness to their parents); overall physical health or stress-related physical problems (e.g., sleep or digestive problems, headaches); and academic or cognitive outcomes (e.g., grades, attentiveness in class, and tests of cognitive development).

## Results

**Comparison of the empirical literature with the theoretical hypotheses.** We compared the results from the studies against the four specific hypotheses presented in Section 1 (see Table 3 for the full results). Under the instability hypothesis, there were some studies that showed equality between SPC and LPC whereas other studies did not explicitly compare the two custody arrangements. In the table, we have separated the studies that made explicit comparisons from those that did not, under each hypothesis. Further, under the fewer resources hypothesis, we have further split studies that show no significant differences in children's outcomes between nuclear families and SPC and studies which show that children in nuclear families have better outcomes compared to children in SPC. We have organized the table in this fashion to aid quick readability of the main findings.

Most studies supported the fewer resources hypothesis (51.3%) suggesting that children from divorced or separated families have fewer resources than nuclear families. Children living in LPC arrangements have the least access to relational and economic resources. Selection hypothesis and instability hypothesis were supported by around quarter of the studies (23.1% and 25.6% respectively). Selection hypothesis suggests that the effects of separation and custody arrangement will go away if selection variables were accounted for, and the instability hypothesis suggests that divorce and separation are in and of themselves stressful events leading to poorer outcomes for children. There was no support for the stressful mobility hypothesis which suggested that LPC arrangements would be preferable to SPC because children would not need to move around as often. A total of 29 out of the 39 studies (74.4%) suggested that either there is no difference between children's outcomes in nuclear families and SPC arrangements or the differences go away after including certain explanatory variables. In contrast, in as many studies (74.4%) the results showed that children in LPC arrangements showed worse outcomes compared to nuclear families. Finally, 20 studies (51.3%) showed that children's outcomes in SPC arrangements were better than in LPC arrangements but these were not always compared as some studies only compared SPC and LPC against nuclear families.

**Do the results differ based on which outcomes were examined in the studies?.** We have split the results according to the outcome variables in Table 3. Emotional outcomes were the

**Table 2. List of the included studies.**

| # | Authors | Year | Dataset[a] | Sample Size | Children's Ages | Data Type | Representative | Country |
|---|---|---|---|---|---|---|---|---|
| 1 | Bacro & de Medeiros | 2020 [12] | | 64 | 3–6 y/o | Cross-Sectional | No | France |
| 2 | Barumandzadeh et al. | 2016 [49] | | 2017 | 11–12 y/o | Cross-sectional | Yes | France |
| 3 | Bastaits et al. [a] | 2012 [22] | Divorce in Flanders | 587 | 10–18 y/o | Cross-Sectional | Yes | Belgium |
| 4 | Bastaits et al. | 2014 [45] | Divorce in Flanders | 684 | 10–18 y/o | Cross-Sectional | Yes | Belgium |
| 5 | Bergström et al. | 2014 [4] | NordChild | 1297 | 4–18 y/o | Cross-Sectional | Yes | Sweden |
| 6 | Bergström et al. | 2019 [8] | NordChild | 3662 | 2–9 y/o | Cross-Sectional | Yes | 5 Nordic countries |
| 7 | Bergström et al. | 2013 [3] | National Survey of Wellbeing and Mental Health | 164580 | 12&15 y/o | Cross-Sectional | Yes | Sweden |
| 8 | Bergström et al. | 2018 [50] | | 3656 | 3–5 y/o | Cross-Sectional | No | Sweden |
| 9 | Bergström et al. | 2015 [7] | National Survey of Wellbeing and Mental Health | 147839 | 14–17 y/o | Cross-Sectional | Yes | Sweden |
| 10 | Bergström et al. | 2021 [51] | | 12845 | 3 y/o | Cross-Sectional | Yes | Sweden |
| 11 | Bjarnason et al. | 2012 [52] | HBSC | 184496 | 11,13&15 y/o | Cross-Sectional | Yes | 36 Western countries |
| 12 | Carlsund et al. | 2013a [53] | HBSC | 11294 | 11–15 y/o | Cross-Sectional | Yes | Sweden |
| 13 | Carlsund et al. | 2013b [54] | HBSC | 3699 | 15 y/o | Cross-Sectional | Yes | Sweden |
| 14 | Chau et al. | 2017 [55] | | 1559 | 10–16 y/o | Cross-Sectional | Yes | France |
| 15 | Cyr et al. | 2013 [56] | | 1414 | 8 y/o | Cross-Sectional | Yes | Canada |
| 16 | Dinisman et al. | 2017 [57] | | 20343 | 10–12 y/o | Cross-Sectional | Yes | 10 countries |
| 17 | Dissing et al. | 2017 [58] | | 44509 | 10–14 y/o | Cross-Sectional | Yes | Denmark |
| 18 | Dujeu et al. | 2022 [59] | | 39294 | 10–19 y/o | Cross-Sectional | Yes | Belgium |
| 19 | Eurenius et al. | 2019 [60] | | 7179 | 3 y/o | Cross-Sectional | Yes | Sweden |
| 20 | Fransson et al. | 2016 [28] | ULF | 4684 | 10–18 y/o | Cross-Sectional | Yes | Sweden |
| 21 | Fransson et al. | 2018 [9] | ULF | 5000 | 10–18 y/o | Cross-Sectional | Yes | Sweden |
| 22 | Hagquist | 2016 [61] | | 172298 | 12&15 y/o | Cross-Sectional | Yes | Sweden |
| 23 | Hjern et al. | 2021a [62] | Danish National Birth Cohort | 31519 | 11 y/o | Longitudinal | Yes | Denmark |
| 24 | Hjern et al. | 2021b [63] | | 30795 | 3 y/o | Cross-Sectional | Yes | Sweden |
| 25 | Hjern et al. | 2021c [64] | Danish National Birth Cohort | 39661 | 7 y/o | Longitudinal | Yes | Denmark |
| 26 | Hoffmann | 2017 [65] | HBSC | 193202 | 11,13&15 y/o | Cross-Sectional | Yes | 36 Western countries |

*(Continued)*

**Table 2.** (Continued)

| # | Authors | Year | Dataset[a] | Sample Size | Children's Ages | Data Type | Representative | Country |
|---|---------|------|------------|-------------|-----------------|-----------|----------------|---------|
| 27 | Kalmijn | 2016 [66] | | 873 | 14 y/o | Cross-Sectional | Yes | Netherlands |
| 28 | Khlat et al. | 2020 [67] | | 39115 | 17 y/o | Cross-sectional | Yes | France |
| 29 | Laursen et al. | 2019 [68] | | 219226 | 9–16 y/o | Cross-Sectional | Yes | Denmark |
| 30 | Låftman et al. | 2014 [14] | | 8840 | 15–16 y/o | Cross-Sectional | Yes | Sweden |
| 31 | Neoh & Mellor | 2010 [69] | | 94 | 8–15 y/o | Cross-Sectional | No | Australia |
| 32 | Nilsen et al. | 2018 [70] | youth@hordaland | 7707 | 16–19 y/o | Cross-Sectional | Yes | Norway |
| 33 | Nilsen et al. | 2020 [71] | youth@hordaland | 7683 | 16–19 y/o | Cross-Sectional | Yes | Norway |
| 34 | Nilsen et al. | 2022a [72] | youth@hordaland | 8833 | 16–19 y/o | Cross-Sectional | Yes | Norway |
| 35 | Nilsen et al. | 2022b [73] | youth@hordaland | 7914 | 16–19 y/o | Cross-Sectional | Yes | Norway |
| 36 | Svensson & Johnson | 2022 [74] | | 3838 | 14–15 y/o | Cross-Sectional | Yes | Sweden |
| 37 | Turunen et al. | 2017 [36] | ULF | 4823 | 10–18 y/o | Cross-Sectional | Yes | Sweden |
| 38 | Turunen et al. | 2021 [75] | | 11802 | 11–15 y/o | Cross-Sectional | Yes | Sweden |
| 39 | Wadsby et al. | 2014 [76] | | 3076 | 18 y/o | Cross-Sectional | Yes | Sweden |

a. Dataset is only mentioned if there are multiple studies using the same data.

most commonly examined across the studies followed by behavioral and then relational and physical outcomes. Educational outcomes were only included in two of the studies and thus are not discussed further.

The fewer resources hypothesis was the most commonly supported hypothesis for emotional and physical outcomes suggesting that children in LPC arrangements have the least resources and thus the lowest emotional and physical outcomes. Half the time (48.0%) children in SPC families also showed worse emotional outcomes and 70% of the time worse physical outcomes compared to nuclear families suggesting that there may be something about certain SPC families or contexts that result in fewer resources and thus poorer children's emotional and physical outcomes.

The results were split between instability hypothesis and fewer resources hypothesis for behavioral and relational outcomes. The results suggest that children following union dissolution have worse behavioral and relational outcomes, some of which can be explained by these children having fewer resources. Some of the studies only compared SPC and LPC against nuclear families rather than comparing between the custody arrangements. Of the studies which directly compared SPC and LPC, 45.5% of the time children in LPC arrangements had worse behavioral and 62.5% of the time worse relational outcomes compared to SPC families.

**Are the results better explained by certain explanatory variables?.** A range of explanatory variables were used across the studies, which were divided into the following variables:

**Table 3.** Systematic review results for each outcome.

| Authors | Year | Outcome | | | | | Explanatory variables |
|---|---|---|---|---|---|---|---|
| | | **Emotional** | **Behavioral** | **Relational** | **Physical** | **Educational** | |
| | | **(n = 25)** | **(n = 16)** | **(n = 11)** | **(n = 10)** | **(n = 2)** | |
| **Selection hypothesis: NF = SPC = LPC 9 (23.1%)** | | **6 (24.0%)** | **3 (18.8%)** | **1 (9.1%)** | **2 (20.0%)** | **0 (%)** | |
| Bastaits et al. | 2012 | NF = SPC≥LPC | | | | | Sample n<1000 SPC = LPC when including relationship with child variables in the analyses. |
| Bastaits et al. | 2014 | NF = SPC = LPC | | | | | Sample n<1000 No explanatory variables were included. |
| Bergström et al. | 2021 | NF = SPC≥LPC | | | | | Sample n>10,000. No change in the results when child or selection variables are included. NF<SPC = LPC when controlling for coparenting quality. |
| Carlsund et al. | 2013b | | NF≥SPC = LPC | | | | Several outcome variables. NF = SPC for sexual debut and conduct problems. Economic and relationship with child variables included in all models. |
| Cyr et al. | 2013 | NF = SPC SPC = LPC, NF>LPC | NF = SPC, SPC = LPC, NF>LPC | | | | No explanatory variables. NF>SPC for hyperactivity. |
| Dissing et al. | 2017 | NF≥SPC≥LPC | | | | | Sample n>10,000. Selection variables controlled for in some analyses. If satisfied with living arrangement: NF = SPC = LPC. No change when include stepparent. |
| Dujeu et al. | 2022 | NF≥SPC≥LPC | | | NF≥SPC≥LPC | | Sample n>10,000. Child, economic, and relationship variables controlled for in some analyses. NF = SPC = LPC when communication with parents included. |
| Fransson et al. | 2018 | | NF = SPC≥LPC | NF = SPC≥LPC | | | Child, economic, parent, and combination of control variables included in different analyses. No change. |
| Turunen et al. | 2021 | | | | NF≥SPC≥LPC | | Child and selection variables controlled for in different analyses. Difference between living arrangements reduces but remains significant. |
| **Instability hypothesis: NF>SPC, LPC 10 (25.6%)** | | **7 (28.0%)** | **7 (43.8%)** | **5 (45.5%)** | **2 (20.0%)** | **1 (50.0%)** | |
| Bergström et al. | 2014 | NF>SPC, LPC | NF>SPC, LPC | NF>SPC, LPC | | | Child and economic variables controlled for in some analyses. |
| Bergström et al. | 2013 | NF>SPC, LPC | NF>SPC, LPC | NF>SPC, LPC | | | Sample n>10,000. Selection and child variables controlled for in some analyses. A few analyses NS for 15y/o (all significant for 12y/o): NF = SPC, LPC for autonomy, peer-relations, and social acceptance. |
| Chau et al. | 2022 | NF>SPC, LPC | NF>SPC, LPC | | NF>SPC, LPC | NF>SPC, LPC | Child gender and age controlled for in the analyses. No change when include stepparent. |
| Hoffmann | 2017 | | NF>SPC, LPC | | | | Sample n>10,000. Economic variables controlled for. |

*(Continued)*

**Table 3.** (Continued)

| Authors | Year | Outcome | | | | | Explanatory variables |
|---------|------|---------|---|---|---|---|------------------------|
| | | Emotional | Behavioral | Relational | Physical | Educational | |
| | | (n = 25) | (n = 16) | (n = 11) | (n = 10) | (n = 2) | |
| Laursen et al. | 2019 | | | NF>SPC, LPC | | | Sample n>10,000. Child and selection variables controlled for in different analyses. No change when include stepparent. |
| *SPC = LPC* | | *4 (57.1%)* | *3 (42.9%)* | *2 (40.0%)* | *1 (50.0%)* | | |
| Carlsund et al. | 2013a | NF>SPC = LPC | | | NF>SPC = LPC | | Sample n>10,000. Child, economic, and relationship with child variables controlled for in different analyses. |
| Hjern et al. | 2021a | NF>SPC≥LPC | NF>SPC≥LPC | NF>SPC≥LPC | | | Sample n>10,000. Child, parent, and selection variables controlled for in different analyses. SPC = LPC when selection variables controlled for. Having a stepparent similar to LPC. |
| Hjern et al. | 2021c | NF>SPC = LPC | NF>SPC = LPC | NF>SPC = LPC | | | Sample n>10,000. Child gender, parent, maternal satisfaction with father, and selection variables controlled for. After controls, stepfamilies no longer worse than SPC/LPC. |
| Kalmijn | 2016 | NF>SPC = LPC | | | | | Sample n<1000. Economic variables controlled for in the analyses. |
| Khlat et al. | 2020 | | NF>SPC≥LPC | | | | Sample n>10,000. Economic variables controlled for. SPC = LPC (mother) for binge drinking. Having a stepparent similar to LPC. |
| **Fewer resources hypothesis: NF≥SPC>LPC 20 (51.3%)** | | **12 (48.0%)** | **6 (37.5%)** | **5 (45.5%)** | **6 (60.0%)** | **1 (50.0%)** | |
| *NF>SPC 6 (30.0%)* | | *4 (33.3%)* | *1 (16.7%)* | *1 (20.0%)* | *2 (33.3%)* | *0 (0.0%)* | |
| Barumandzadeh et al. | 2016 | NF>SPC>LPC | | | | | Parent relationship variables controlled for. |
| Bergström et al. | 2019 | NF>SPC>LPC | NF>SPC>LPC | NF>SPC>LPC | | | Child, country, and economic variables controlled for. |
| Bergström et al. | 2015 | | | | NF>SPC>LPC | | Sample n>10,000. Child, economic, relationship with child, and combination of all controlled for in different analyses. Differences get smaller but remain significant for each model. SPC = LPC for boys when all variables are controlled for. |
| Bjarnason et al. | 2012 | NF>SPC>LPC | | | | | Sample n>10,000. Child, economic, and parent relationship variables controlled for. Having a stepparent equal to LPC. |
| Låftman et al. | 2014 | | | | NF>SPC>LPC | | Child, relationship with child, economic and interactions between custody and relationship with child controlled for in different analyses. Families with stepparents become NS when economic variables and interactions are controlled for. |
| Neoh & Mellor | 2010 | NF>SPC>LPC | | | | | Sample n<100. Selection variables controlled for. |
| *NF = SPC 14 (66.7%)* | | *8 (66.7%)* | *5 (83.3%)* | *4 (80.0%)* | *4 (66.7%)* | *1 (100.0%)* | |
| Bacro & de Medeiros | 2021 | | NF = SPC>LPC | NF = SPC>LPC | | | Sample n<100. Parent relationship controlled for. |

*(Continued)*

**Table 3.** (Continued)

| Authors | Year | Outcome | | | | | Explanatory variables |
|---|---|---|---|---|---|---|---|
| | | Emotional | Behavioral | Relational | Physical | Educational | |
| | | (n = 25) | (n = 16) | (n = 11) | (n = 10) | (n = 2) | |
| Bergström et al. | 2018 | NF = SPC>LPC | NF = SPC>LPC | NF = SPC>LPC | | | Child and selection variables controlled for. |
| Eurenius et al. | 2019 | NF = SPC>LPC | | NF = SPC>LPC | | | Child gender controlled for. |
| Fransson et al. | 2016 | NF = SPC>LPC | | | | | Child, economic, parent, and combination of all covariates controlled for in different analyses. |
| Fransson et al. | 2018 | NF = SPC>LPC | | | NF = SPC>LPC | | Child and economic variables controlled for in different analyses. SPC = LPC for psychological complaints. SPC = LPC for psychological complaints and emotional well-being when controlling for economic variables. SPC = LPC for alcohol use. SPC = LPC for weekly exercise when economic variables controlled for. Most analyses remained significant. |
| Hagquist | 2016 | | | | NF≥SPC>LPC | | Sample n>10,000. Selection, economic and relationship variables controlled for. The differences become smaller when accounting for more variables. IF = SPC for 9[th] graders when relationship accounted for. |
| Hjern et al. | 2021b | NF = SPC>LPC | NF = SPC>LPC | | | | Selection variables controlled for. |
| Nilsen et al. | 2018 | NF = SPC>LPC | NF = SPC>LPC | | | | Controlling for economic variables makes the difference between SPC and LPC smaller but remains significant. Having a stepparent same as LPC. |
| Nilsen et al. | 2020 | | | | NF = SPC>LPC | | Sibling relationships controlled for. Having a stepparent same as LPC. |
| Nilsen et al. | 2022a | | | | NF = SPC≥LPC | | No explanatory variables were included. Not all outcomes were significant. Having a stepparent same as LPC. |
| Nilsen et al. | 2022b | | | | | NF≥SPC>LPC | Selection, economic, relationship, and sibling relationship variables controlled for. NF>SPC significant when economic variables controlled for, otherwise NF = SPC. |
| Svensson & Johnson | 2022 | | NF = SPC>LPC | | | | Relationship with child variables controlled for. Having a stepparent same as LPC. |
| Turunen et al. | 2017 | NF≥SPC≥LPC | | | | | Sample n>10,000. NF = SPC for getting enough sleep, bedtime, and social jetlag. NF>SPC for sleep initiation difficulties. |
| Wadsby et al. | 2014 | NF = SPC>LPC | | NF = SPC>LPC | | | No explanatory variables were included. |
| **Stressful mobility hypothesis: NF≥LPC>SPC 0 (0.0%)** | | | | | | | |

*Note.* NF = nuclear family, SPC = shared physical custody, LPC = lone physical custody. NS = non-significant. If confidence intervals available or there is a comparison between all custody arrangements, the custody arrangements are sorted in order, otherwise the comparison is just with the reference category. ≥ indicates that the effect is sometimes significant, sometimes not. Samples with exceptionally small (n<100 or n<1000) or large (n>10,000) are denoted separately. In addition to the studies being divided into the four hypotheses, we have also provided additional information in bold italics where there are two different options within the hypothesis (i.e., rows denoted with SPC = LPC, NF = SPC and NF>SPC). The percentages in these subcategories are based on the overall number of studies supporting the specific hypothesis rather than the overall number of studies.

- Selection (parent demographic variables; n = 14),

- Relationships (e.g., parenting, parents' relationship quality; n = 15),

- Economic (e.g., education, occupation, income, n = 17),

- Child (e.g., age, gender, n = 20), and

- Having a stepparent (n = 12).

Five studies included no explanatory variables. In most cases, the inclusion of explanatory variables did not alter the significance of the results. The most common explanatory variables that made the results non-significant was relationship variables (n = 3) followed by child's age (n = 2; sometimes at least some of the outcomes were not significantly different for older children). Selection and economic variables made the results non-significant only in one study each. However, in most studies the inclusion of explanatory variables did not change the significance of the results although in many studies it reduced the differences in the outcomes between the three groups. Children living in stepfamilies reported outcomes that were similar to children in LPC arrangements.

**Methodological findings.** In depth methodological critique of the existing literature has been provided elsewhere [21]. Thus, we only provide a brief description of the methodological considerations of the current studies.

Only two of the 39 (5.1%) studies reviewed included multiple waves of data in their analyses but neither of the studies examined effects longitudinally over time. These two studies [62, 64] found that children's outcomes were better in nuclear families compared to SPC or LPC whereas SPC and LPC were equal. We did find several longitudinal studies on separation and divorce during our review process, but these studies did not include a comparison of custody arrangements and were thus beyond the scope of this review.

The majority of the studies included large samples with only five (12.8%) studies including less than 1,000 participants and 16 (41.0%) including more than 10,000 participants. Large samples sizes generally allow for more accurate conclusions to be drawn. However, they are also likely to render very small but clinically meaningless differences significant. Two (40.0%) of the studies with less than 1,000 participants and three (18.8%) studies with more than 10,000 participants found no significant differences between custody arrangements. Thus, studies with larger samples were more likely to find significant differences between the different living arrangements.

Interestingly, although only a small number of studies which included exclusively young children (0–7 years), all studies except one [64] showed that children in nuclear families and SPC families had equal outcomes [12, 50, 51, 60, 63]. In contrast, children in LPC families had the worst outcomes in four of the six (60.0%) studies.

## Discussion

The purpose of the present systematic review was to examine whether there are systematic differences in children's outcomes across family living arrangements (nuclear family, SPC, LPC). We also aimed to establish which of the four theoretical hypotheses (selection, instability, fewer resources, and stressful mobility hypothesis) were supported by the research findings. Finally, we examined whether there were specific explanatory variables that were more likely to account for differences between family living arrangements. There were two main conclusions regarding differences between family living arrangements and the theoretical hypotheses that could be drawn from the review. First, in line with the research showing that divorce has a negative impact on children [2, 3, 5, 8, 14], children in nuclear families consistently showed

equal or better outcomes compared to post-separation living arrangements. There were no studies in which children in nuclear families had worse outcomes than children in other family arrangements. This finding is consistent with the fewer resources and instability hypotheses. However, in 75% of the studies children in SPC did equally well compared to children in nuclear families. This suggests that it is not the instability resulting from divorce or separation per se that leads to worse outcomes for children. In many cases, it is possible to achieve comparable outcomes for children if parents are able to agree to and successfully have an SPC arrangement. Therefore, the results are the most consistent with the fewer resources hypothesis which suggests that children in LPC arrangements have fewer economic and relationship resources compared to children in SPC arrangements and especially in nuclear families.

Second, the only theoretical hypothesis that was not supported by the literature was the stressful mobility hypothesis, which suggests that SPC arrangements are worse for children compared to LPC arrangements because of the stress induced by moving between households or the potential ruptures in primary caregiving relationships due to moving. This is in line with a review by Nielsen (2018) [6] who showed that overall children in SPC do better compared to children in LPC. This suggests that maintaining connection with both parents outweighs any potential drawbacks of moving between households and supports many countries' policies around preference for SPC over LPC [36, 77].

Additionally, we examined several potential explanatory variables that may explain differences between nuclear families, SPC, and LPC. Including child, selection, relationship, or economic explanatory variables in the analyses often reduced the differences between the groups; yet differences remained significant in most analyses. However, it is not possible to quantify how much an inclusion of the explanatory variables lessened the differences between the three groups because different effect sizes were used across studies and were often unstandardized. Studies with larger sample sizes were more likely to find significant differences suggesting that the overall differences between the studies may have been practically less important. Studies with younger children (0–7 years) suggested that the outcomes between nuclear families and SPC were equal which contrasts with some researchers and practitioners who suggest that younger children are more likely to benefit from a primary caregiver, often the mother [78].

Finally, we also examined whether having a stepparent changed the outcomes for children. Some theories and research findings suggest that the introduction of stepparents or stepfamilies would put additional strain on the family system and thus result in lower children's outcomes [31, 33, 34] but this was not supported by the present systematic review. In general, the outcomes in stepfamilies were equal to outcomes in LPC arrangements [14, 52, 58, 62, 64, 67, 68, 70–72, 74]. A prior systematic review showed that children in stepfamilies tended to have lower well-being compared to children in intact families [79] to which we added results to suggest that stepfamilies are no worse than other post-separation living arrangements. However, it is important to note that none of the studies distinguished between stepfamilies where biological parents were in LPC versus SPC. Therefore, we do not know exactly what the contribution of stepfamilies is in this equation, which is an important avenue for future research.

## Implications for theory, research, and practice

The present systematic review has several key implications for theory and highlights additional questions. The findings suggest that having a stable nuclear family environment is generally better for children's outcomes compared to not having one. This finding supports the instability hypothesis which suggests that union dissolution itself is stressful for children and families. However, children in SPC arrangements had comparable outcomes to children in nuclear families in three fourth of the time for at least some analyses, suggesting that maintaining

connection with both parents following separation or divorce can buffer against the potential negative impact of union dissolution. There was no consistency in which variables made the differences non-significant, however, none of the studies included data from pre-divorce or separation; thus, we do not know if children's outcomes were already poor prior the union dissolution or only following the dissolution (no baseline). The majority of the studies provided support for the fewer resources hypothesis, suggesting that children especially in LPC arrangements have fewer relationship and economic resources compared to children in nuclear and SPC families. However, controlling for relationship and economic variables only rarely made this association go away which suggests that there is something about LPC arrangements that is not determined by explanatory variables that were included in the analyses.

Furthermore, the potential effects of living arrangements on children's outcomes may be conceptualized differently depending on the theoretical perspective adopted in research, yet most of the studies were atheoretical. Only 9 out of 39 (23.1%) studies explicitly referred to one or more theories or theoretical hypotheses to be tested. Overall, the studies reviewed referred to common psychological models such as attachment theory or family systems theory, as well as more specific perspectives, such as the modeling perspective, the escape from stress perspective, or the parental loss theory. The sociological literature included classical theories about the general effects of parental union dissolution on children: family instability, social capital, economic deprivation and strain, and life course or ecological theories. Other studies often referred to specific variables that have been examined in the past literature but did not provide an explicit theoretical justification for why these factors were important. A lack of a solid theoretical foundation leaves researchers a great deal of degrees of freedom making it difficult to interpret the results and to make definitive conclusions from the existing evidence. Therefore, it is imperative that future research makes theoretical assumptions clear and explicit and justifies the inclusion of their variables and groups to help readers situate the findings in the broader literature.

Finally, by providing a solid empirical basis on living arrangements, determinants, and dynamics, the review aimed to provide evidence-based grounding of the necessary adaptation for social policies and legal procedures, which are nowadays still largely based on the nuclear family model and the relatively undefined concept of the "best interests of the child" [80]. This situation is problematic because the societal and legal debate on the best interest of the child is too often informed by ideological positions rather than research [81]. The results suggest that children in nuclear families still do better in general compared to children following union dissolution. Thus, supporting high risk families should be the first focus of policies given a stable nuclear family is likely to provide the most supportive environment for children to grow. This may include, for example, preventive measures such as family and gender policies which facilitate a more equal share of household and childcare tasks (sustainable parental leave policies, encouraging gender-equal pay at work). In cases where a couple chooses to end their relationship, the results suggests that policies should support families to choose and maintain an SPC arrangement and in many cases, it is possible to achieve comparable outcomes to nuclear families. This could be achieved, for example, by discouraging long and conflictual divorce proceedings and instead encouraging shorter and consensual separation procedures which center the best needs of the child(ren).

In addition to policy changes, providing interventions for couples to address their coparenting relationship can help improve children's outcomes. For example, parents' participation in a co-parenting program has been shown to improve overall family well-being [82]. In these programs, parents receive psychoeducation on the importance of their relationship quality for their child, practice communication skills, and are supported in developing co-parenting plans. These programs are designed for parents both before and after separation or divorce [83].

## Strengths, limitations, and future directions

The present systematic review has several strengths. The study of the impact of divorce and separation on children's outcomes is inherently interdisciplinary and the review pulls together research from across different disciplines allowing researchers to consider results of the empirical research and draw conclusions beyond their own discipline. We also summarized and grounded the review in theories present in this area and compared the results against different theoretical predictions. Furthermore, we systematically contrasted nuclear, SPC, and LPC families allowing us to make comparisons between these three groups.

However, there are also several limitations both in the review itself and in the field more generally that should be considered when interpreting the findings. First, there was little consistency in explanatory or outcome variables across the studies making it difficult for us to fulfill our third objective of identifying which explanatory variables make the association between living arrangement and children's outcomes non-significant. This limitation also relates to the lack of theoretical grounding in the research conducted to date. Often there was no justification of why certain explanatory variables were chosen, or why certain outcome variables were included and not others. The explanatory variables should be carefully selected to be able to differentiate between the hypotheses. For example, whether differences in living arrangements are due to selection factors prior to divorce, the parental economic situation, or the current functioning of the parental relationship, each require a different set of explanatory variables to test the hypotheses. Further, educational outcomes were only included in two studies, which meant that there was not sufficient evidence to make conclusions about these outcomes. It is important that future research incorporate testing and comparing theoretical hypotheses and make their theoretical assumptions clear at the start of the investigation.

Second, only two of the 39 studies reviewed included longitudinal data and even in these studies, each of the waves were analyzed separately rather than looking at trajectories over time or controlling for previous timepoints. Therefore, we currently have no data available about children's outcomes in different custody arrangements or stability of custody arrangements over time. For example, SPC may be difficult to maintain because parents may move further away from each other, get into new relationships and have new families, or parents may not be able to maintain large enough homes due to financial strain. Additionally, the longitudinal studies in this research domain tend to include a small number of waves spaced six months or a year apart and there are no studies that we are aware of which examine the daily lives of families across different living arrangements using intensive methods such as momentary sampling, daily diary designs, or weekly follow-ups. Thus, future longitudinal and more intensive research is urgently needed to better understand the trajectories of these arrangements over time and the children's life as it is lived in the different living arrangements respectively.

Finally, we chose to compare the results of various studies across three living arrangements: nuclear, SPC, and LPC families. However, some studies included other groups such as overnight stays or legal custody. These groups did not appear across many studies and thus we could not compare them. In the future it would be important to establish, for example, how much contact with both parents is needed for children to show better outcomes. One study did show that children in families who had a mostly LPC arrangement but maintained contact with the other parent compared to children who were solely in LPC arrangement showed better outcomes [36]. Additionally, there are other forms of family diversity besides custody that we did not consider but would be important to do so in future research. Acknowledging that an increasingly large number of children no longer live primarily in traditional two-biological-parent households and understanding better when and what kind of support may be needed in certain families to help children flourish is important.

## Conclusion

Several main messages can be drawn from the existing literature on the association between living arrangements and children's well-being: (1) children in nuclear families do better than children in various post-separation living arrangements; however, different explanatory variables change the association between living arrangements and children's outcomes in inconsistent ways; (2) children in SPC arrangements are consistently found to do better than children in LPC arrangements and to do equally well compared to children in nuclear families; (3) children in stepfamilies show equal outcomes compared to LPC. In conclusion, the findings from the existing literature provide research-based evidence to shape social policies and legal procedures related to family living arrangements and children well-being.

## Supporting information

**S1 Checklist. PRISMA 2009 checklist.**
(DOC)

## Author Contributions

**Conceptualization:** Laura M. Vowels, Chiara L. Comolli, Laura Bernardi, Daniela Chacón-Mendoza, Joëlle Darwiche.

**Formal analysis:** Laura M. Vowels.

**Methodology:** Laura M. Vowels, Chiara L. Comolli, Laura Bernardi, Daniela Chacón-Mendoza, Joëlle Darwiche.

**Project administration:** Laura M. Vowels, Daniela Chacón-Mendoza.

**Writing – original draft:** Laura M. Vowels.

**Writing – review & editing:** Laura M. Vowels, Chiara L. Comolli, Laura Bernardi, Daniela Chacón-Mendoza, Joëlle Darwiche.

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
