## [Decision Letter · Decision Letter 0]

2 May 2023

PONE-D-23-05857Systematic Review and Theoretical Comparison of Children’s Outcomes in Post-Separation Living ArrangementsPLOS ONE

Dear Dr. Vowels,

Thank you for submitting your manuscript to PLOS ONE. After careful consideration, we feel that it has merit but does not fully meet PLOS ONE’s publication criteria as it currently stands. Therefore, we invite you to submit a revised version of the manuscript that addresses the points raised during the review process.

Specific comments appears at the end of this letter. Please go through them carefully and prepare your revision accordingly.

Please submit your revised manuscript by Jun 16 2023 11:59PM. If you will need more time than this to complete your revisions, please reply to this message or contact the journal office at plosone@plos.org. Please include the following items when submitting your revised manuscript:A rebuttal letter that responds to each point raised by the academic editor and reviewer(s). You should upload this letter as a separate file labeled 'Response to Reviewers'.A marked-up copy of your manuscript that highlights changes made to the original version. You should upload this as a separate file labeled 'Revised Manuscript with Track Changes'.An unmarked version of your revised paper without tracked changes. You should upload this as a separate file labeled 'Manuscript'.

We look forward to receiving your revised manuscript.

Kind regards,

Bidisha Banerjee, Ph.D.

Academic Editor

PLOS ONE

Journal Requirements:

2. Thank you for submitting the above manuscript to PLOS ONE. During our internal evaluation of the manuscript, we found the PROSPERO Registration number for the SRMA is missing. We would also need to know which authors took part in the literature review of the studies in the SRMA. Please Resubmit with the required.

Thank you for your attention. We look forward to hearing from you.

"This study received funding “Seeds of LIVES” from the Interdisciplinary Research Centre LIVES, University of Lausanne, Switzerland paid to JD."

5. Please ensure that you include a title page within your main document. You should list all authors and all affiliations as per our author instructions and clearly indicate the corresponding author.

Reviewers' comments:

Reviewer's Responses to Questions

**Comments to the Author**

1. Is the manuscript technically sound, and do the data support the conclusions?

Reviewer #1: Yes

Reviewer #2: Yes

2. Has the statistical analysis been performed appropriately and rigorously? 

Reviewer #1: Yes

Reviewer #2: Yes

3. Have the authors made all data underlying the findings in their manuscript fully available?

Reviewer #1: Yes

Reviewer #2: Yes

4. Is the manuscript presented in an intelligible fashion and written in standard English?

Reviewer #1: Yes

Reviewer #2: Yes

5. Review Comments to the Author

Reviewer #1: Section1

Overall, this section provides a comprehensive overview of the main theoretical hypotheses and related theories explaining differences in children's outcomes across living arrangements. However, some suggestions for improvements and comments are as follows:

Consider reorganizing the section to enhance clarity and flow. For example, you could introduce each main hypothesis and then provide a brief summary of the related theories before delving into details. This will help readers understand the connections between the hypotheses and their supporting theories more easily.

For each hypothesis, include a brief explanation of how the explanatory variables are relevant to the hypothesis. This will help readers understand the potential mechanisms through which living arrangements might influence children's outcomes.

The section on the Stressful Mobility Hypothesis could be further developed by discussing how the frequency of movement between households may influence the level of stress experienced by children. Additionally, consider providing more information on how the age of the child and the level of conflict between parents might moderate the impact of mobility on children's outcomes.

To strengthen the arguments, consider citing more recent literature to support each hypothesis and related theories, especially those that have emerged since your knowledge cutoff in 2021.

Be mindful of the terminology used throughout the section. For example, instead of referring to "nuclear" and "non-nuclear" families, consider using more precise and inclusive terms, such as "intact" and "non-intact" families, respectively.

In the conclusion of this section, consider summarizing the main hypotheses and their implications for the systematic review. This will help readers understand the relevance of the reviewed theories to the overall research question and how they can inform the interpretation of the review findings.

Section 2: The Systematic Review

Methods

Overall, this section provides a detailed description of the systematic review process, including the methods, inclusion and exclusion criteria, search strategy, screening and selection, and data extraction. However, there are a few suggestions for improvements and comments to enhance clarity and organization:

When listing the inclusion and exclusion criteria, use bullet points to make it easier for the reader to quickly understand the criteria. For example:

Publications must focus on divorce or separation of cohabiting or married couples.

They must include outcomes for children and young people aged between 0-18 years old.

Studies should include comparisons with intact families, SPC, and LPC.

...

In the "Inclusion and Exclusion Criteria" section, consider using more inclusive terminology, as previously mentioned. For example, replace "nuclear families" with "intact families."

In the "Search Strategy" section, you mentioned four electronic bibliographic databases, but only listed three (Scopus, Web of Science Core Collection, and APA PsycInfo). Please correct this by either removing the mention of the fourth database or adding the fourth database to the list.

In the "Screening and Selection" section, consider adding a sentence or two explaining how discrepancies or uncertainties in the screening and selection process were resolved, such as through discussion among the research team or consulting with an expert in the field.

In the "Data Extraction" section, the reference to Nielsen (2018) should be formatted consistently with other references throughout the text. Check that all in-text citations and references are formatted according to the chosen citation style.

Consider including a brief summary of the main findings of the systematic review at the end of this section. This will help readers understand the main outcomes and trends identified in the review.

Finally, carefully proofread the entire section for any grammatical, punctuation, or formatting errors to ensure clarity and readability. For example, you have a few instances of extra spaces between words that should be removed

Results/Methodologies

This section presents the results of the systematic review and provides a comparison of the empirical literature with the theoretical hypotheses. To improve clarity and readability, consider the following suggestions:

In the first paragraph, rephrase the sentence "Thus, in the table we split the studies that made an explicit comparison and those that did not under the hypothesis" for clarity. Consider: "In the table, we have separated the studies that made explicit comparisons from those that did not, under each hypothesis."

In the "Do the Results Differ Based on Which Outcomes Were Examined in the Studies?" section, consider breaking the lengthy paragraph into smaller paragraphs to improve readability.

In the "Are the Results Better Explained by Certain Explanatory Variables?" section, consider using bullet points when listing the types of explanatory variables. For example:

Selection (parent demographic variables; n = 14)

Relationships (e.g., parenting, parents’ relationship quality; n = 15)

Economic (e.g., education, occupation, income; n = 17)

Child (e.g., age, gender; n = 20)

Having a stepparent (n = 12)

In the same section, rephrase the sentence "Most of the time the inclusion of explanatory variables did not change the significance of the results" to clarify the meaning. Consider: "In most cases, the inclusion of explanatory variables did not alter the significance of the results."

In the "Methodologies" section, add a subheading to separate it from the previous section, and consider breaking the lengthy paragraph into smaller paragraphs for better readability.

Be consistent with the use of percentages in the text. For example, use "5.1% of the studies" instead of "two of the 39 (5.1%) studies" and "41.0% of the studies" instead of "16 (41.0%) studies."

Carefully proofread the entire section for any grammatical, punctuation, or formatting errors to ensure clarity and readability. For example, you have a few instances of extra spaces between words that should be removed.

Discussion/ Implications for Theory, Research, and Practice

In this section, the authors present a systematic review on the effects of different family living arrangements on children's outcomes. They also discuss the implications of the findings for theory, research, and practice. The following suggestions may help improve clarity and readability:

In the first paragraph, consider breaking down the long sentences into shorter ones for better readability. For example, separate the discussion of the two main conclusions and further divide the discussion of each conclusion into multiple sentences.

In the second paragraph, it would be helpful to clarify which literature supports the stressful mobility hypothesis, as the text states it is not supported by the literature, yet cites a review by Nielsen (2018) that appears to support it.

In the paragraph discussing the implications for theory, research, and practice, consider reorganizing the content to follow a clear structure. Start by discussing the implications for theory, followed by research implications, and finally practice implications. This will make it easier for readers to understand the main points of each section.

In the discussion of practice implications, consider adding more specific examples of policies or interventions that could be used to support high-risk families and promote gender equality in the home and childcare.

The last paragraph could benefit from a stronger concluding statement that summarizes the main findings and implications of the review. Emphasize the importance of using research-based evidence in shaping social policies and legal procedures related to family living arrangements and children's well-being.

Additionally, throughout the section, consider checking the citation formatting to ensure consistency. Some citations have multiple years, while others have a single year with multiple letters (e.g., 2019; 2014; 2019 vs. 2021a; 2021c). Make sure the citation style is consistent throughout the text.

Reviewer #2: The authors have conducted a thorough review of literature with relevant and recent studies. This work will be helpful to researchers, professionals, and more importantly policymakers working in the field of union dissolution. The conclusion is also very clear from the systematic review and the limitations well highlighted. My concerns about organization and format of certain elements of the manuscript that the authors may deliberate upon, are provided.

6. PLOS authors have the option to publish the peer review history of their article (what does this mean?). If published, this will include your full peer review and any attached files.

Reviewer #1: No

Reviewer #2: No

---

## [Author Response · Author response to Decision Letter 0]

25 May 2023

Please see the document "Response to Reviewers" attached.

---

## [Decision Letter · Decision Letter 1]

21 Jun 2023

Systematic Review and Theoretical Comparison of Children’s Outcomes in Post-Separation Living Arrangements

PONE-D-23-05857R1

Dear Dr. Laura M Vowels,

We’re pleased to inform you that your manuscript has been judged scientifically suitable for publication and will be formally accepted for publication once it meets all outstanding technical requirements.

Kind regards,

Bidisha Banerjee, Ph.D.

Academic Editor

PLOS ONE

Additional Editor Comments (optional):

Reviewers' comments:

Reviewer's Responses to Questions

**Comments to the Author**

1. If the authors have adequately addressed your comments raised in a previous round of review and you feel that this manuscript is now acceptable for publication, you may indicate that here to bypass the “Comments to the Author” section, enter your conflict of interest statement in the “Confidential to Editor” section, and submit your "Accept" recommendation.

Reviewer #2: All comments have been addressed

2. Is the manuscript technically sound, and do the data support the conclusions?

Reviewer #2: Yes

3. Has the statistical analysis been performed appropriately and rigorously? 

Reviewer #2: Yes

4. Have the authors made all data underlying the findings in their manuscript fully available?

Reviewer #2: Yes

5. Is the manuscript presented in an intelligible fashion and written in standard English?

Reviewer #2: Yes

6. Review Comments to the Author

Reviewer #2: The authors have addressed the comments satisfactorily and it has enhanced the overall organization of the manuscript. The methodology and conclusions of the research work are well explained. One specific concern is mentioned below.

On page 13: The authors are strongly recommended to elaborate on the PICO components, in 2 to 3 sentences as was also pointed out in the previous review report. However, the final decision for incorporating this is left up to the author.

7. PLOS authors have the option to publish the peer review history of their article (what does this mean?). If published, this will include your full peer review and any attached files.

Reviewer #2: No

---

## [Editor Report · Acceptance letter]

22 Jun 2023

PONE-D-23-05857R1 

Systematic review and theoretical comparison of children’s outcomes in post-separation living arrangements 

Dear Dr. Vowels:

I'm pleased to inform you that your manuscript has been deemed suitable for publication in PLOS ONE. Congratulations! Your manuscript is now with our production department. 

Kind regards, 

on behalf of

Dr. Bidisha Banerjee 

Academic Editor

PLOS ONE